# Reinforced Concrete Structures Containing Chopped Carbon Fibers with Polymer Composite Materials

**DOI:** 10.3390/polym13213812

**Published:** 2021-11-04

**Authors:** George Soupionis, Loukas Zoumpoulakis

**Affiliations:** Department of Chemical Engineering, Zografou Campus, National Technical University of Athens, 9 Heroon Polytechniou Str., 15773 Athens, Greece; lzoubou@chemeng.ntua.gr

**Keywords:** composite materials, carbon fibers, reinforced concrete, glass fiber, Kevlar^®^ 49

## Abstract

In this paper, the reinforcement of concrete combining composite materials with carbon, glass and aramid fibers is discussed. Accordingly, cement specimens reinforced with chopped carbon fibers were manufactured via the retrofitting method and coated using various different types of fabrics (carbon, glass and aramid), and epoxy resin systems were developed and studied using compressive strength tests. In addition, polymeric matrix (epoxy resin) composite materials reinforced with different types of fabric (carbon, glass and aramid (Kevlar^®^ 49)) were manufactured and their shear and bending strengths were measured. Before reinforcing cement specimens, all fabrics (carbon, glass and aramid (Kevlar^®^ 49)) were placed in a vacuum chamber and were processed via pre-impregnation. This specific reinforcing method significantly improved the mechanical properties of cementitious structures with compressive strength values that reached 81 MPa. In a similar way, the bending and shear strengths of the materials under study were measured at 405 MPa and 33 MPa, respectively.

## 1. Introduction

Since the 1960s, the construction industry has taken steps to adopt fiber-reinforced polymer composites (FRPs) in various technological aspects. As a result of these efforts, FRPs have been applied successfully in load-bearing and panel filling applications, pressure pipes, tank lining and roof construction. In recent years, FRP composite structures have been developed and used in many complex constructions and infrastructure works including bridges, roads and rail networks [1,2,3,4].

The characteristic usability and flexibility of these materials allows for their application in structural, decorative and local repair projects. The superior mechanical properties and low weight that characterize the majority of composite materials has attracted the interest of engineers in the research and development of FRP materials that are appropriate for utilization in the construction industry [5,6,7,8,9,10,11,12,13]. FRPs were successfully applied for the first time during the 1990s in the reinforcement of concrete columns [8]. As a result, many similar emerging issues in the repair and strengthening of various other existing structures were placed under scientific and technical investigation [9,10].

As a reinforcing material used in concrete structures, FRPs exhibit remarkably high mechanical properties and are also characterized by outstanding plasticity. However, their compression strength appears to be approximately 20% to 50% lower in comparison to their shear strength [14,15,16,17], which constitutes the main load that these structures are being subjected to. Materials of this category provide high levels of resistance when used in corrosive environments and high stress applications, contributing to the minimization of long-term repair costs while at the same time maintaining structural integrity (in terms of statics) [18,19,20,21]. Extensive research using a great variety of fabrics (glass, carbon, aramid and basalt) was carried out on the improvement in strength and durability of FRP-reinforced beams, columns, walls and other structural parts of buildings [22,23,24,25,26,27]. These fabrics with enhanced mechanical properties appear to be appropriate for use in demanding structural applications for which high performance and low weight are prerequisites [28,29,30].

In addition, in order to prevent damage caused by corrosion phenomena as a result of environmental conditions (humidity, temperature variation, air pollution, negative ions, etc.) and considering that composites are quite time-durable compared to other materials, such as wood, metal and bricks, that become vulnerable when exposed to such conditions, polymer matrix composite systems are often used as protection [31,32,33,34,35]. These materials are also known for their adequate thermal and electrical insulation efficiency [36,37], a very important feature in construction applications, since currently the issue of building energy efficiency has become crucial.

However, such materials are highly flexible when thermal or electrical conductivity is required, since they may modify into conductors when specific additives are used in their manufacture. These properties can be combined into composite materials [38,39,40].

Despite all the advantageous factors discussed above, no significant amount of research has been carried out in this specific field.

The novelty of this study is that it introduces new concrete reinforcing materials based on the use of chopped carbon fibers with polymer composite materials [41]. The great variety of advantageous characteristics of these new reinforcing patterns is attributed to the combination of carbon fibers and polymer composite materials that results in materials of increased mechanical performance. The scope of this research is to investigate the mechanical characteristics of these new composites and provide some conclusions on their adequacy for use in reinforcement applications.

In the present work, cement specimens containing chopped carbon fibers, wrapped using fabrics of carbon fiber, glass fiber and aramid fiber with an epoxy resin matrix, were manufactured according to the standard EN 196-1:1995 [42,43]. Additionally, composites using epoxy resin as a matrix and carbon, glass and aramid fiber fabrics as reinforcements were also made. The mechanical properties of all of the resulting materials and their specific flexural and shear strength were measured.

## 2. Materials and Methods

Two different types of epoxy resin systems purchased from the market were used. Analytically, the epoxy paste-type resin system (Sika, Swiss), a high-strength resin mainly applied in fabric reinforcements, and the liquid epoxy resin system (SINTECNO, Hellas), a low-viscosity solvent-free resin (WINTER: 140 ± 28 mPa s/SUMMER:320 ± 64 mPa s) suitable for reinforcing fabrics were used in the manufacture of composite specimens. In order to achieve better adhesion of the applied reinforcement, an epoxy resin base (SINTECNO, Hellas) was used as a primer.

Two types of carbon fiber fabric were used for the retrofitting reinforcement. One-directional fabrics and two-directional (twill) as shown in Figure 1a,b. As far as the glass fiber fabric and the Kevlar fabric are concerned, the two-directional (twill) type fabric was exclusively used (Figure 2a,b). Although this specific twill fabric is used as a reinforcement in concrete structures and has not been widely referred to in the literature, it was chosen because of its loose weave pattern that allows better application in complex shapes (Figure 1c). While the one-directional carbon fiber fabric has a mono-fiber number of 3000 and a density of 1.80 g/cm^3^, the two-directional carbon, glass and aramid fiber fabrics are twill knit. The carbon fiber fabric has a mono-fiber number of 3000 and a density of 1.80 g/cm^3^. The glass fiber fabric had an areal weight of 0.028 g/cm², and the aramid fabric (Kevlar 49) weighed 0.03 g/cm^2^. For the cement specimens’ reinforcement, the chopped carbon fiber used was 0.60 cm and had a density of 1.80 g/cm^3^.

The cement specimens were manufactured according the standard EN 196-1, with Portland-type cement and sand, with the following proportions: cement, 450 g (±2); sand, 1350 g (±5); and water, 225 g (±1). The cement mold based on the EN 196-1 standard is presented in Figure 2a. After curing, the cement specimens were placed in water for 28 days (Figure 2b). In the case of the reinforced samples, 1.70 g of chopped carbon fibers were added (10% *w*/*w* of the cured samples). Subsequently, when the drying process was complete and considering that the edges of concrete structures receive the largest part of the applied load (tension/pressure), the surfaces and edges were appropriately treated (surface smoothing and edge rounding) to enable the resin’s optimum adhesion as well as the optimum application of carbon, glass and aramid fabrics (Figure 3a). To improve the adherence of fabrics to the surface of the cement specimens, the defective locations of the specimens were treated using special paste epoxy resin (SINTECNO Hellas) with additives (quartz powder).

When the surface treatment procedure was complete, the resulting specimens were covered with the composite fabric reinforcing material. Prior to wrapping, all cloths were placed in a vacuum chamber (Figure 4a,b) and were pre-impregnated in a 200 mL solution of 132 g acetone, 45.30 g resin and 22.70 g hardener.

The specimens that were wrapped with the composite material using liquid resin as a matrix were treated with a primer for better adhesion of the cloth to the cement surface (Figure 5b). The cement specimens were reinforced peripherally with two layers of the composite material on all four sides (Figure 6a). The retrofitting method was chosen in order to increase the shear strength of the concrete columns, enhance their ductility and improve the confinement level of the concrete during mechanical stress [1,2].

The one-directional carbon fiber fabric that reinforced the cement specimen was placed vertically to the compression force. The fibers of the two-dimensional fabric were vertical in one direction and parallel to the compression force in the other, as illustrated in Figure 1a,b. In addition, the paste epoxy resin was easier to apply during the manufacturing process.

The composites were manufactured with epoxy resin with unidimensional and two-dimensional (twill) fabrics to measure their shear and bending strength. All composites were made using the hand lay-up method [7]. Figure 6b illustrates the dimensions of the samples for the measurement of shear and bending strength.

The calculation of the layers of fabrics for the manufacture of the composite materials was in accordance with the thickness of the fabrics. The thickness for the unidimensional carbon fabric was 0.630 mm and for the two-dimensional (twill) fabrics, the thicknesses were 0.285 mm for carbon, 0.300 mm for glass, and 0.400 mm for Kevlar. Hence, to manufacture composite materials with 3 mm thickness, the unidimensional carbon fabric required five layers weighing 358 g. In terms of the two-dimensional (twill) fabrics, for carbon, eleven layers were required weighing 405 g; for glass, ten layers were needed weighing 324 g; and for Kevlar, eight layers weighing 506 g were required (according to ASTM D-7617). Additionally, to manufacture the composite materials, the fabric/resin ratio was 50% *w*/*w*. The volume ratio of fabric/resin was 53% *v/v*. For the reinforced cement specimens with the retrofitting method, the fabric volume ratio was 3% *v/v* (for two layers of fabric). For the cement specimens reinforced with chopped carbon fibers, the fiber/cement ratio was 0.1% *w/w*. The specimens made in the present study are depicted in Figure 7, Figure 8 and Figure 9, while the specimens suitable for the bending and shear strength measurements are shown in Figure 10.

For the manufacture of the cement and composite specimens, epoxy resin, unidimensional and two-dimensional carbon, glass and aramid fiber fabrics were used, as shown in Table 1.

## 3. Mechanical Properties

The measurements of the mechanical properties of the composites are presented in Figure 10. More specifically, bending strength and shear strength were measured following the three-point method based on the ASTM D2344-65T (BS EN ISO 14125: 1998), ASTM D7617 and ASTM D790-71 (Figure 11). Based on the standards for general bending tests, the speed should be no higher than 2 mm/min, and the support span should be 16 (tolerance ±) times the thickness of the specimen (in this project, 5 cm). For the shear tests, the speed should be no higher than 2 mm/min and the span should be 8 (tolerance ±) times the thickness (in this project, 2 cm). For both the bending and shear tests, the specimen is placed horizontally on a support span. The loading nose of a special dynamometer applies the load at the center of the specimen and measures the resulting deflection as a proportional indication by applying pressure (1 mm/min). The result links to a force in a table given by the manufacturer. Then, using Equations (1) and (2), the bending strength *σ*_b_ (MPa) and the shear strength *τ*_b_ (MPa) are calculated.
(1)$$\sigma_{b}=\frac{3P_{max}l_{s}}{2bd^{2}},$$
(2)$$\tau_{b}=0.75\frac{P_{max}}{bd}$$

***P***_max_: maximum load, as the specimen breaks (N), ***l_s_***: test length (m), ***b***: width (m) and ***d***: thickness (m).

The measurement of the compressive strength of the cement specimens wrapped in carbon, glass and Kevlar fabrics and epoxy resin as a matrix was carried out following the EN 196-1:1995 standard. The tests were performed using a MATEST TREVIOLO 24048 (made in Italy) press of capacity 2000 kN (Figure 12). The results are in kN, and the compressive strength is calculated *σ* (MPa), according to Equation (3).
(3)$$\sigma_{\theta }=\frac{F}{A}$$

***F***: maximum load when the specimen breaks (kN); ***A***: specimen surface area in contact with the press (mm^2^).

## 4. Results

The specimens tested in the bending and shear strength tests are presented in Figure 13. In the shear strength test, the specimens with carbon fiber and glass fiber fabric (all cases of each category) broke completely. In contrast, those tested for bending strength (all specimens) did bend, but did not break. The reason for this is that in bending strength, the fabric plays a crucial role as it receives the full load and prevents the specimen from breaking.

Figure 14 shows the shear strength results, while Figure 15 shows the bending strength results of the composite materials reinforced with one-directional and two-directional fabrics. As shown in Figure 14, the shear strength in all cases ranges at 30 MPa, with the lowest strength found in the Kevlar sample, which reached 21 MPa. The specimens reinforced with one-directional fabrics exhibited a slightly higher shear strength (33 MPa). It is clear from Figure 15 that the composite materials with one-directional carbon fabrics outperformed the two-directional carbon fabrics in bending strength by ~30%. However, regarding the glass fiber fabric (Gp), the difference was ~3%. As for the matrix, the resin exhibiting better bending strength was liquid resin, reaching 405 Mpa (Caodl).

As shown in Figure 16, Figure 17 and Figure 18, the breaking in all specimens examined [17,18] was parallel to the direction of the applied force, with the fracture point appearing at the edges of the specimens where the weakest points of the structure are located. Throughout the measurements, in all specimens, most main rupture incidents occurred at the edge of the composite material.

Figure 19 shows the compressive strength measurement results for cementitious composites and cementitious composites reinforced with chopped carbon fibers, respectively. As can be clearly observed, the compressive strength of cement specimens reinforced with chopped carbon fibers and wrapped with the two-directional carbon fabric and paste resin (Ccatp) increased by ~44%, reaching a value of 81 MPa, compared to the specimens without retrofitting reinforcement. For all other cases studied, the compressive strength increased by ~20%. All of the paste-type resin matrix specimens appeared to perform better (in terms of compressive strength), exhibiting increased durability in comparison to all of the other materials developed and tested. According to the results (Figure 19), the addition of chopped carbon fibers increased the compressive strength of the resulting composites by ~14%. The chopped carbon fibers that they are in the concrete samples are visible in Figure 20a,b.

## 5. Conclusions

This work was an experimental, technical and theoretical study of the behavior of small-scale cement beams reinforced with FRP composite materials in compression strength. The main conclusions drawn are as follows:As reinforcement, the carbon fiber fabrics exhibited better results in all cases, independent of the form of the epoxy matrix.The liquid epoxy resin as a matrix worked better with carbon fiber fabrics than the other materials due to the resin’s improved penetration into the fibers. The maximum bending and shear strength appeared to be present in the composite material reinforced with unidimensional carbon fiber fabric.The investigation revealed that the shear strength of all composite materials did not change remarkably. The composite materials reinforced with carbon fabrics were only slightly higher in the scale regarding strength.Increasing compressive strength of 44% was evident in the case of a cement specimen reinforced with a paste epoxy resin composite material with a two-dimensional carbon fiber fabric (Ccatp). Moreover, the twill pattern fabrics generated far better results.It was observed that the paste epoxy resin was easier to apply to the cement specimens. However, it must be noted that the two-dimensional (twill) carbon fiber fabric adhered better to the specimen.The addition of chopped carbon fibers into the cement increased compressive strength to a notable degree.The Kevlar fabric showed difficulty in absorbing the resin during pre-pregnation. It is noted that during the measurements, the samples appeared to be broken at certain points where the fabric looked dry. In contrast, the glass fabric demonstrated excellent absorption of all types of resin and exhibited satisfactory results in the measurements, especially in bending strength.

This study reveals that the best option to reinforce cement structures is carbon fibers and, according to the results, the investigated materials are suitable for this specific purpose. The composite material that produces better results in reinforcing cementitious specimens with the retrofitting method is the paste resin as a matrix and two-directional (twill) fabric as a reinforcement that reaches the value of 81 MPa. Furthermore, the addition of chopped carbon fibers into the cement provides better strength in all cases of external reinforcement.

## Figures and Tables

**Figure 1 polymers-13-03812-f001:**
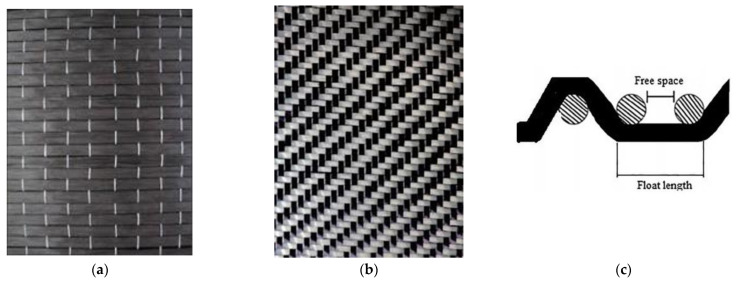
Carbon fiber fabric: (**a**) Unidimensional; (**b**) two dimensional twill; (**c**) weaving of the twill pattern.

**Figure 2 polymers-13-03812-f002:**
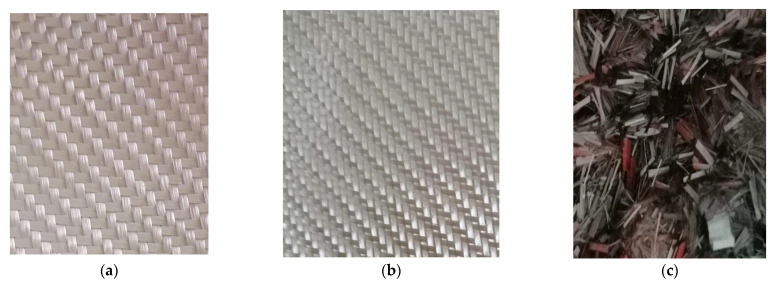
Fiber cloth: (**a**) Kevlar^®^ 49 twill fabric; (**b**) glass twill fabric; (**c**) chopped carbon fiber 0.6 cm.

**Figure 3 polymers-13-03812-f003:**
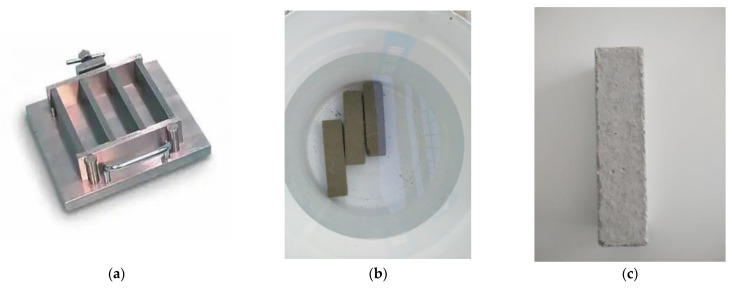
(**a**) Three-gang mold for 40 × 40 × 160 mm cement specimens; (**b**) 28-days hydration of cement specimens; (**c**) cement specimen.

**Figure 4 polymers-13-03812-f004:**
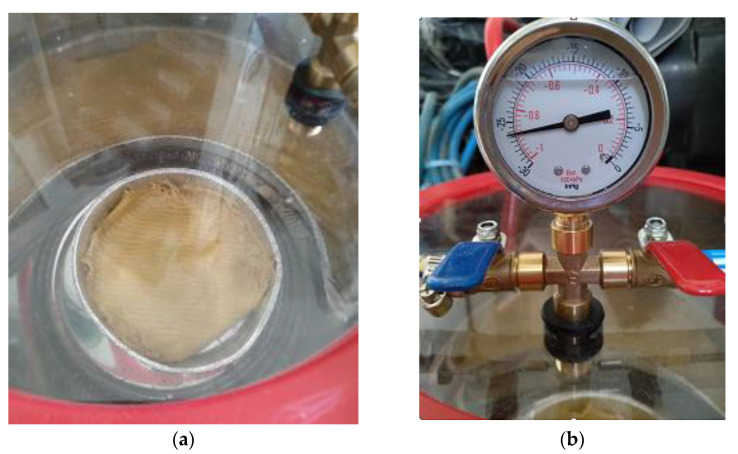
(**a**) Pre-impregnation of Kevlar cloth; (**b**) vacuum gauge.

**Figure 5 polymers-13-03812-f005:**
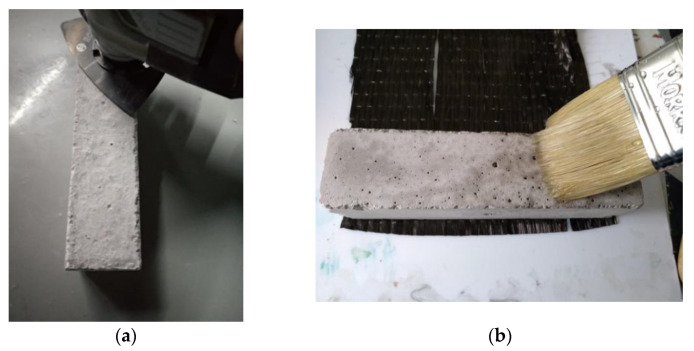
(**a**) Method of smoothing the specimen surfaces; (**b**) priming the surfaces for better adhesion of the fabric.

**Figure 6 polymers-13-03812-f006:**
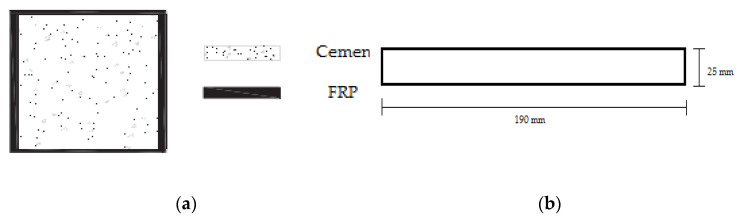
(**a**) Cement specimens reinforced peripherally with CFRP; (**b**) specimens’ dimensions for the bending and shear strength measurement.

**Figure 7 polymers-13-03812-f007:**
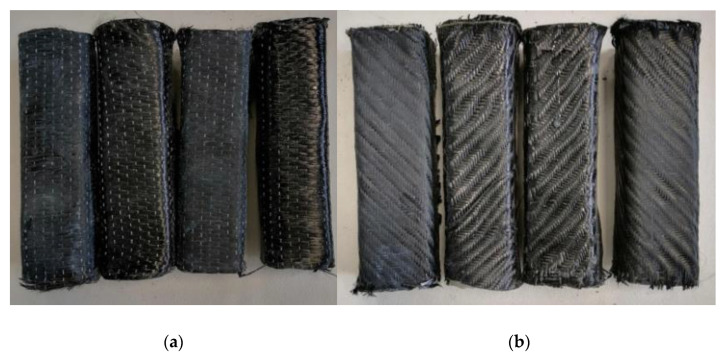
All carbon fiber reinforcement cases of cement specimens: (**a**) case of unidimensional fabric; (**b**) case of twill fabric.

**Figure 8 polymers-13-03812-f008:**
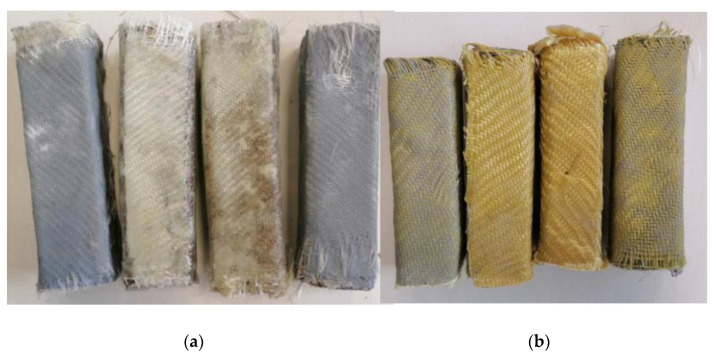
Glass fiber and Kevlar reinforcement cases of cement specimens: (**a**) glass fiber fabric; (**b**) Kevlar fabric.

**Figure 9 polymers-13-03812-f009:**
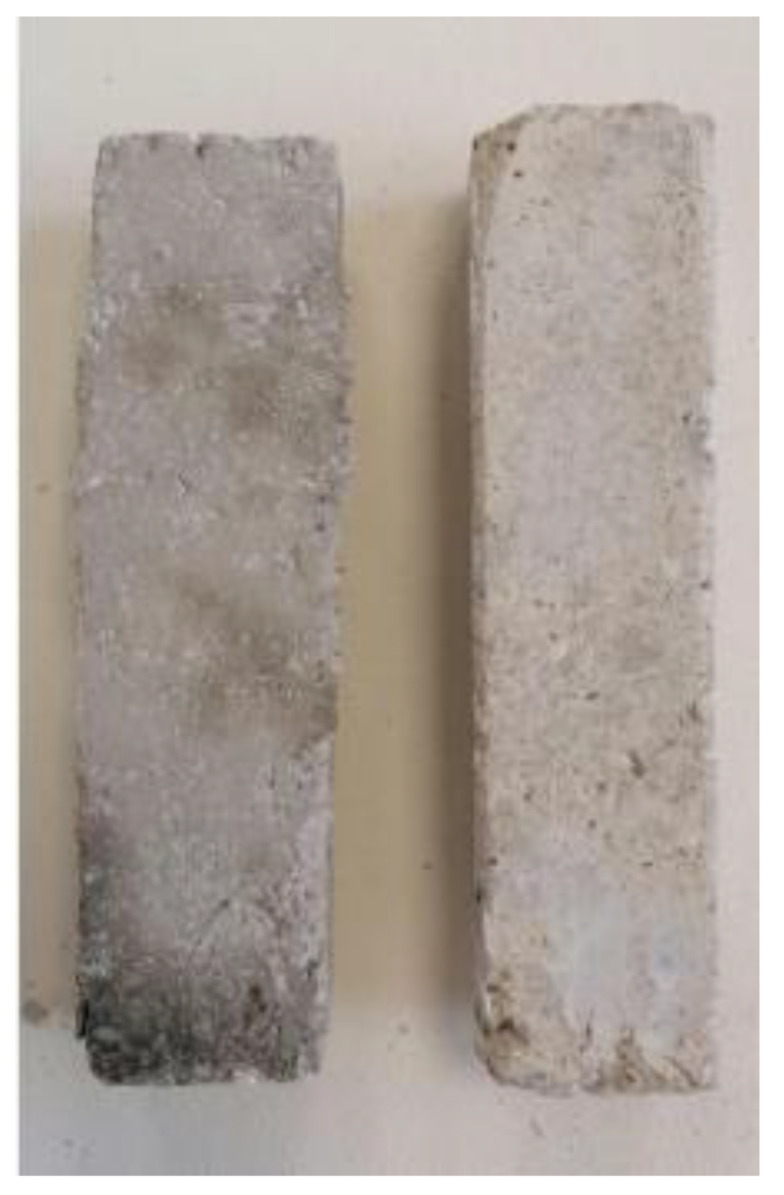
Cement specimens without retrofitting reinforcement.

**Figure 10 polymers-13-03812-f010:**
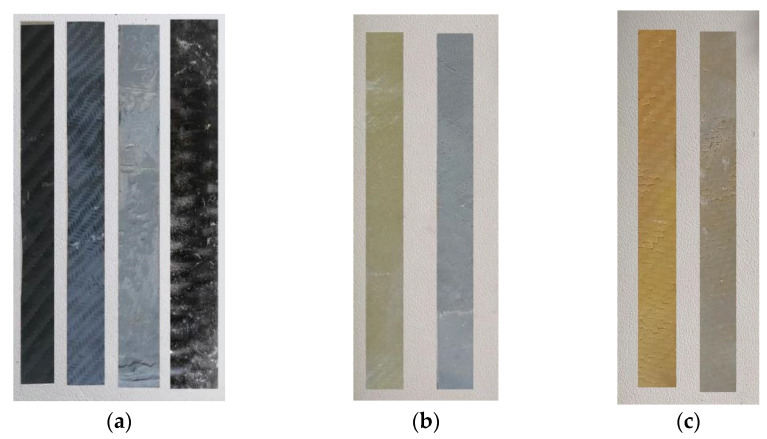
Composite material specimens for the shear and bending strength measurements: (**a**) carbon fiber-reinforced; (**b**) glass fiber-reinforced; (**c**) Kevlar-reinforced.

**Figure 11 polymers-13-03812-f011:**
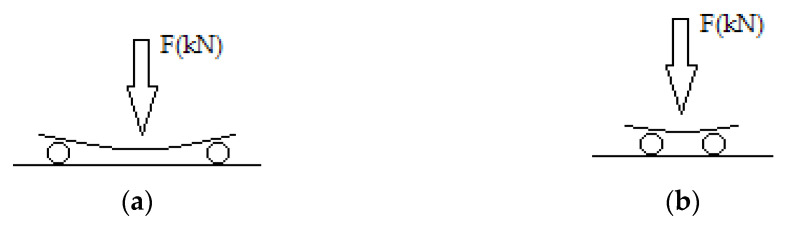
(**a**) Three-point scheme for the bending strength measurement; (**b**) three-point scheme for the shear strength measurement.

**Figure 12 polymers-13-03812-f012:**
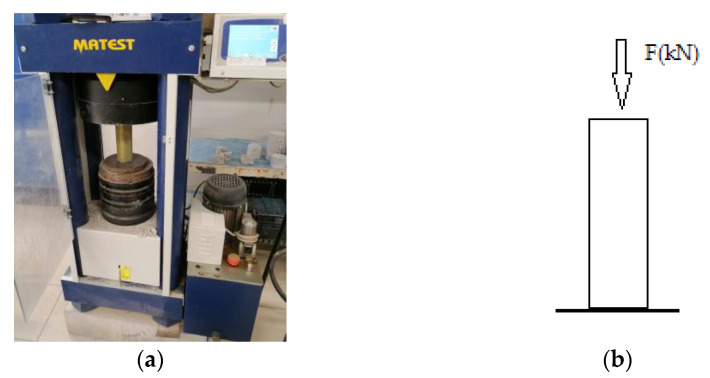
(**a**) MATEST TREVIOLO press of capacity 2000 kN during the compression tests; (**b**) representation of compression measurement.

**Figure 13 polymers-13-03812-f013:**
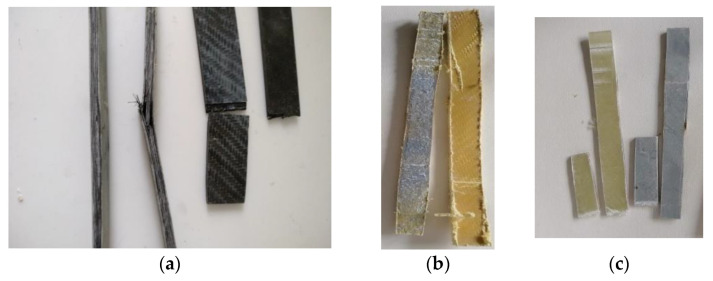
Specimens tested in bending and shear strength: (**a**) carbon fiber specimens; (**b**) Kevlar specimens; (**c**) glass fiber specimens.

**Figure 14 polymers-13-03812-f014:**
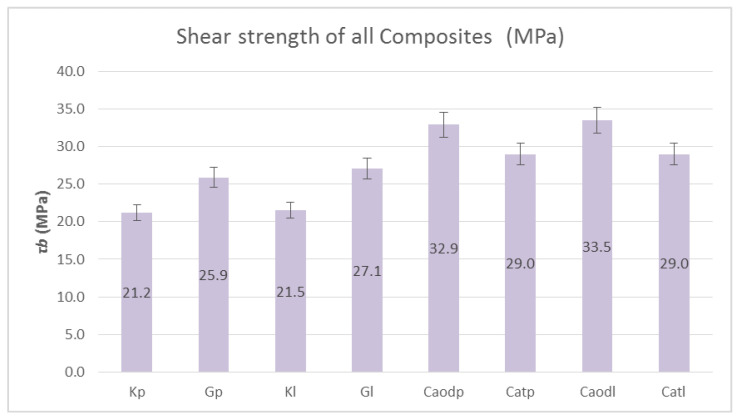
Shear strength of epoxy resin-reinforced specimens (reinforced with carbon fiber, glass fiber and Kevlar fabrics). Variation of values: ±5%.

**Figure 15 polymers-13-03812-f015:**
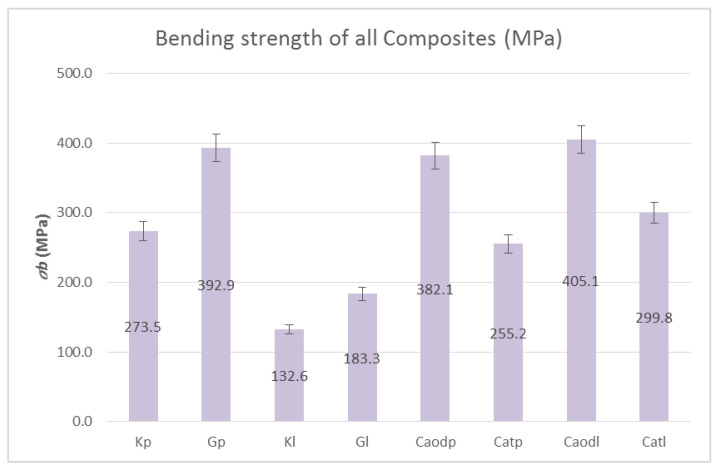
Bending strength of epoxy resin-reinforced specimens (reinforced with carbon fiber, glass fiber and Kevlar fabrics). Variation of values: ±5%.

**Figure 16 polymers-13-03812-f016:**
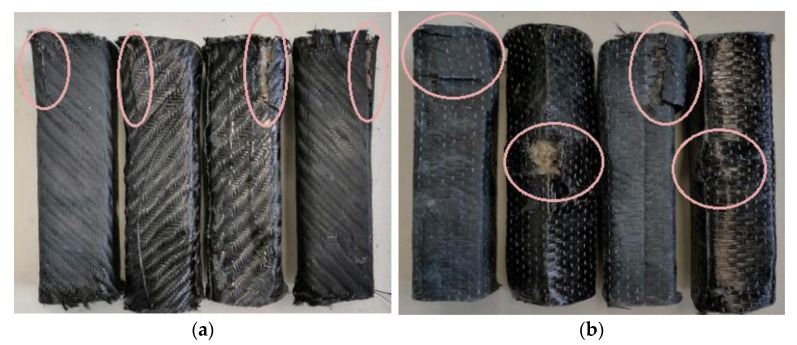
Compressed specimens reinforced with carbon fabric: (**a**) unidimensional fabric; (**b**) twill fabric.

**Figure 17 polymers-13-03812-f017:**
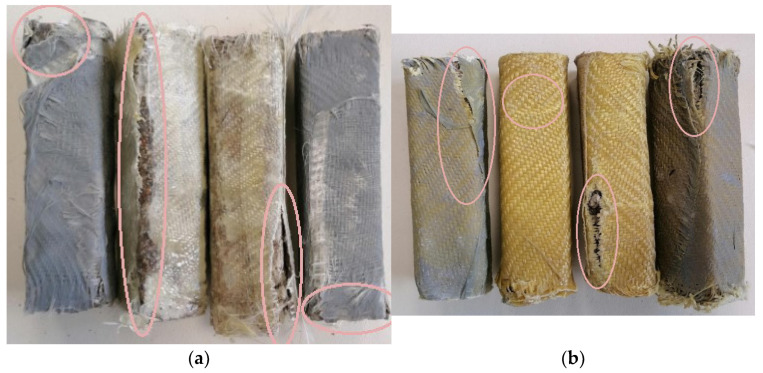
Compressed specimens reinforced with glass fiber fabric and Kevlar: (**a**) glass fiber (**b**) Kevlar fabric.

**Figure 18 polymers-13-03812-f018:**
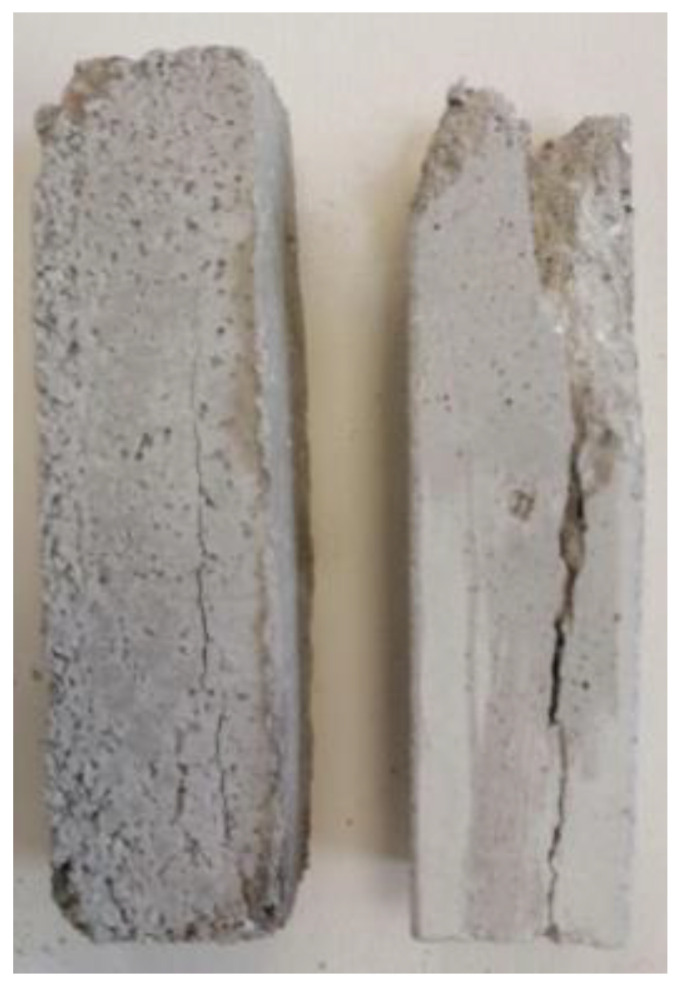
Compressed cement specimen without reinforcement (retrofitting).

**Figure 19 polymers-13-03812-f019:**
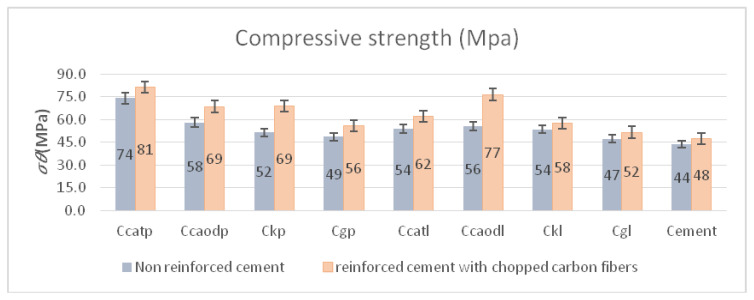
Compressive strength of cement specimens reinforced with chopped carbon fibers and non-reinforced specimens wrapped with composite materials of epoxy resin as a matrix and one-directional and two-directional carbon fiber fabric, glass fiber fabric and Kevlar for reinforcement. Variation of values: ±5%.

**Figure 20 polymers-13-03812-f020:**
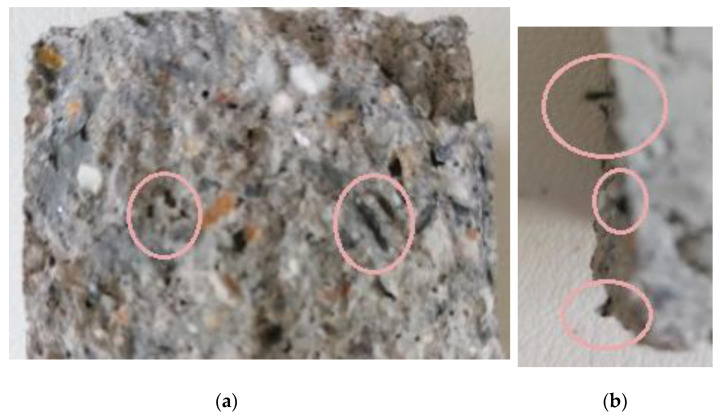
Compressed cement specimen with chopped carbon fiber reinforcement (**a**) Front view of broken sample (**b**) side view of broken sample.

**Table 1 polymers-13-03812-t001:** Composite materials of epoxy resin–carbon fiber and cement specimens.

Composite Material Code	Cement (% *w*/*w*)	Carbon Fibers (% *v*/*v*)			Glass (% *v*/*v*)	Kevlar (% *v*/*v*)	Epoxy Resin (% *w*/*w*)	
		One Direction	Twill	Chopped	Twill	Twill	Liquid	Paste
C	100	0	0	0	0	0	0	0
Rc	90	0	0	10	0	0	0	0
Ccatp	80	0	10	0	0	0	0	10
Ccaodp	80	10	0	0	0	0	0	10
Ckp	80	0	0	0	0	10	0	10
Cgp	80	0	0	0	10	0	0	10
Ccatl	80	0	10	0	0	0	10	0
Ccaodl	80	10	0	0	0	0	10	0
Ckl	80	0	0	0	0	10	10	0
Cgl	80	0	0	0	10	0	10	0
Rccatp	70	0	10	10	0	0	0	10
RCcaodp	70	10	0	10	0	0	0	10
RCkp	70	0	0	10	0	10	0	10
RCgp	70	0	0	10	10	0	0	10
RCcatl	70	0	10	10	0	0	10	0
RCcaodl	70	10	0	10	0	0	10	0
RCkl	70	0	0	10	0	10	10	0
RCgl	70	0	0	10	10	0	10	0
Caodp	0	50	0	0	0	0	0	50
Caodl	0	50	0	0	0	0	50	0
Catp	0	0	50	0	0	0	0	50
Catl	0	0	50	0	0	0	50	0
Kl	0	0	0	0	0	50	50	0
Kp	0	0	0	0	0	50	0	50
Gl	0	0	0	0	50	0	50	0
Gp	0	0	0	0	50	0	0	50

Notes: Od: unidimensional, T: twill (two-dimensional), P: paste, L: liquid, C: cement, Rc: reinforced cement, Ca: carbon, K: Kevlar, G: glass.

## Data Availability

The data presented in the study are available on request from the corresponding author.

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
