# Peer review of "Reinforced Concrete Structures Containing Chopped Carbon Fibers with Polymer Composite Materials"

_polymers, 2021, doi:10.3390/polym13213812_

Round 1

Reviewer 1 Report

The current paper must be made the adequate modifications to improve quality. There are many logic and nonstandard writing problems. It is suggested that the authors fully consider the following comments and make the sufficient modification.

  1. The title is inappropriate because this article uses three kinds of fibers. The title only refers to carbon fiber. Please modify it.
  1. The present writing of abstract is very poor, and it does not include some key contributions, for example the key results or conclusions. It is suggested that the authors rewrite the abstract and add some important conclusions.
  1. The current introduction is too little, and the authors does not provide the enough summary on the research significance. The current writing is too broad and lacks of the specific summary of other related research work. Furthermore, the logical structure of the introduction is also chaotic. Specific comments are as follows:

1) The mechanical properties, fatigue and corrosive resistances of fiber reinforced polymer composites in construction field should be fully summarized and supported by relevant literature. The advantages and performance of composite should be analyzed to provide the important research significance. In addition, the present paper studied carbon fiber, glass fiber and aramid. However, the properties and advantages of the above three fibers have not been summarized. It is suggested that the authors see the latest research on mechanical, fatigue and durability of carbon fiber and glass fiber composite exposed to the temperature, humidity or loading. Journal of Materials Research and Technology, 2021, 14:2812-2831.  Composite Structures, 2021, 255: 112869.  Fatigue & Fracture of Engineering Materials & Structures, 2019; 42: 1148-1160.

2) The mechanical properties of three kinds of fiber reinforced cement samples are studied in the current paper. However, the reviewers did not see the summary on the mechanical properties of fiber reinforced cement materials. For example, the influence law and mechanism of fiber addition on the properties of cement-based materials. It is suggested that the authors fully summarize the relevant research work and gives the significance of the present paper.

3) It is suggested that the authors rewrite the introduction around the key scientific problems to be solved in the present paper. Specifically, it includes the summary on the application background, material, properties and mechanism.

  1. The current writings on the materials and methods are confused. The authors should write this part according to raw materials, sample preparation and performance test. Furthermore, the mechanical property test in part 3 should be integrated into materials and methods.
  1. Almost, each figure and equation in the paper are not clear and standardized. It is suggested that the authors rewrite them in a standardized way.
  1. Table 1 shows the parameters obtained during sample preparation, which should be placed in the material and method section, rather than in the results and discussion section.
  2. Figure 14 and Figure 15 lack of the abscissa and ordinate and the related scale are missing.
  3. No information can be seen from Figure 20 because the quality of the picture is too poor.
  4. The current conclusion is too poorly written and needs to be highly condensed, including key results, rather than the description of the experimental phenomena.

Author Response

I would like to thank you for your valuable suggestions in improving this article. I hope that all the alterations were sufficient in meeting your requirements. As we live in a demanding time due to the COVID pandemic, it is understandable that these experiments were conducted with many difficulties and even in periods where the laboratories were practically closed. It is particularly in this respect that I want to thank you for your help in advancing our research in such difficult times.

Reviewer 2 Report

Abstract

"In all specimens, mechanical properties were measured." Please specify the mechanical test together with the standard follows.

Materials and Methods

Please specify the viscosity of the epoxy resin.

3. Mechanical properties

Authors is advised to discuss further on interfacial bonding between the glass/carbon and Kevlar towards the mechanical properties.

Lack of discussion in effect of c polymeric matrix composite materials reinforced with carbon, glass, and aramid fiber to fortify concrete.

Conclusion:

"The liquid epoxy resin as a matrix works better with carbon fiber fabrics than
the other materials due to the resin's improved penetration into the fibers."Please remove this, since the authors does not compare the penetration with other matrix.

Author Response

(The authors gave the same response as above.)

Round 2

Reviewer 1 Report

Although the authors tried to make a modification to the paper, however, the present modification is not satisfactory. The authors did not provide a point-to-point response, nor did they give the explanation of the questions. In addition, the authors did not make sufficient changes for the introduction according to the comments of the reviewers. The revised introduction is still not enough and does not include some key summary related to this work. It is suggested to consider the first comment to make a full revision and add the necessary summary to the related references.

“The current introduction is too little, and the authors does not provide the enough summary on the research significance. The current writing is too broad and lacks of the specific summary of other related research work. Furthermore, the logical structure of the introduction is also chaotic. Specific comments are as follows:

1) The mechanical properties, fatigue and corrosive resistances of fiber reinforced polymer composites in construction field should be fully summarized and supported by relevant literature. The advantages and performance of composite should be analyzed to provide the important research significance. In addition, the present paper studied carbon fiber, glass fiber and aramid. However, the properties and advantages of the above three fibers have not been summarized. It is suggested that the authors see the latest research on mechanical, fatigue and durability of carbon fiber and glass fiber composite exposed to the temperature, humidity or loading. Journal of Materials Research and Technology, 2021, 14:2812-2831.  Composite Structures, 2021, 255: 112869.  Fatigue & Fracture of Engineering Materials & Structures, 2019; 42: 1148-1160.

2) The mechanical properties of three kinds of fiber reinforced cement samples are studied in the current paper. However, the reviewers did not see the summary on the mechanical properties of fiber reinforced cement materials. For example, the influence law and mechanism of fiber addition on the properties of cement-based materials. It is suggested that the authors fully summarize the relevant research work and gives the significance of the present paper.

3) It is suggested that the authors rewrite the introduction around the key scientific problems to be solved in the present paper. Specifically, it includes the summary on the application background, material, properties and mechanism.”

Reviewer 2 Report

Authors has addressed their manuscript accordingly.

For the last reminder, the authors need to check some unit, e.g. Mpa-> MPa.

Please remove: "Those types of epoxy resins are available in the market."